# A Study on the Properties of Recycled Aggregate Concrete and Its Production Facilities

**Jung-Ho Kim [1],\*** , **Jong-Hyun Sung [1]**, **Chan-Soo Jeon [2]**, **Sae-Hyun Lee [2]** and **Han-Soo Kim [3]**

1   Technology Research Center, Hallaencom, 1170-5, Poseunghyangnamro, Hwaseong-si, Gyeonggi-do 18572, Korea; jonghyun.sung@hallaencom.com

2   Department of Living and Built Environment Research, Korea Institute of Civil Engineering and Building Technology, 283, Goyang-daero, Ilsanseo-gu, Goyang-si, Gyeonggi-do 10223, Korea; jcsi0815@kict.re.kr (C.-S.J.); shlee@kict.re.kr (S.-H.L.)

3   Department of Architecture, Konkuk University, 120 Neungdong-ro, Gwangjin-gu, Seoul 05029, Korea; hskim@konkuk.ac.kr

\*   Correspondence: jungho.kim@hallaencom.com; Tel.: +82-10-5057-5390

**Abstract:** In recent years, the amount of construction waste and recycled aggregate has been increasing every year in Korea. However, as the recycled aggregate is poor quality, it is not used for concrete, and the Korean government has strengthened the quality standards for recycled aggregate for concrete. In this study, research was conducted on the mechanical and durability characteristics of concrete using recycled aggregate, after developing equipment to improve the quality of recycled aggregate to increase the use of recycled aggregate for environmental improvements. The results illustrated improvements in the air volume, slump, compressive strength, freezing and thawing resistance, and drying shrinkage. Furthermore, this study is expected to contribute to the increased use of recycled aggregate in the future.

**Keywords:** recycled aggregate; concrete; construction waste; mechanical characteristics; durable characteristics

## 1. Introduction

In the latter part of the 1980s, Korea constructed large-scale housing developments, as well as new cities for building pleasant residential environments and resolving housing issues. Thereafter, during the 2010s, with the drastic increase of old housing that caused re-development and re-construction to become increasingly active, the production of construction waste materials increased yearly. According to the nationwide waste material volume of 2016, which was reported by the Korea Environment Corporation in 2017. The volume of construction waste materials has been on the rise every year (as shown in Figure 1), with the highest value reaching up to approximately 48%. From the figure, the ratio of waste concrete was 62.8% and the waste asphalt concrete was 17.9%, illustrating that the recycling of waste concrete has become an urgent matter [1].

In accordance with the provision of Article 35 of the Construction Waste Recycling Promotion Act, the government established the Recycled Aggregate Quality Standard for promoting the recycling of waste materials from construction, to recommend a more diversified and broadened facilitation of waste concrete. Therefore, in terms of applying recycled aggregate for concrete, there have been a significant number of studies that have been conducted at home and abroad, and recently, there have been active studies on the addition of diversified mixed materials to the recycled aggregate concrete or the mixed use of natural aggregate, to provide recycled aggregate concrete with capabilities equivalent to that of ordinary concrete [2–14]. However, recycled aggregate for use in concrete has not been widely

employed in Korea, as the aggregates are of poor quality; they have a high absorption rate and low density, therefore, are mostly used for filling the ground.

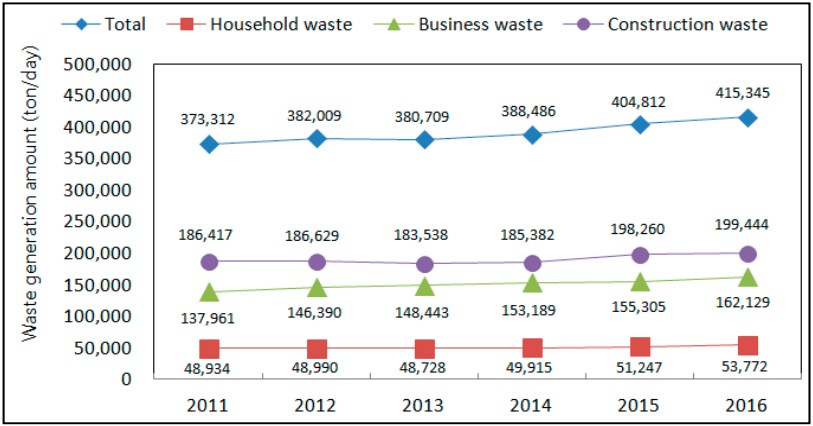

**Figure 1.** Changes in the waste material volume in Korea [1].

The recycled aggregates are not used in structural buildings as their quality fluctuation is severe. It might only be used in buildings if the cement paste on the surface of the recycled aggregate is removed, but the removal of impurities and fine aggregates is also very important. Previous research, has studied the removal of cement paste on the surface of recycled aggregates but there have been no research on the development of impurities and the aggregate fine powder removal facilities. Therefore, the main purpose of this study was to develop and verify facilities, to activate recycled aggregate for concrete. Theoretically, there exists a technology that improves the quality of recycled aggregate by adding wind speed to the impact method, which removes the impurities.

Previous studies that have pursued the improvement of the quality of recycled aggregates, have mostly been conducted on the removal of surface cement paste, and the removal methods were typified by impact, chemical, and heating methods. Ogawa and Nawa reported that the impact strength increased as the number of impacts increased, and that the compressive strength of the recycled aggregate concrete decreased [15]. Shima and Tatayasiki have reported a technique of removing the cement paste on the surface of the recycled aggregate, by heating and fracturing [16]. Kim, Kiuchi, Juan, and Ismail, as a study of chemical methods, focused on the technique of removing the cement paste adhered to the recycled aggregate, with hydrochloric acid solution [17–20]. Tam also reported cement paste removal with sulfuric acid solution [12]. However, the heating and fracturing and chemical methods have not been utilized due to its spatial and temporal constraints.

In order to promote the use of recycled aggregates, the Korean government has presented a new standard of use with an expanded scope that requires the use of recycled fine aggregate, from the existing recycled coarse aggregate. This standard was in accordance with the laws and regulations of the quality standard of the recycled aggregate. In the event that the standard of use for the recycled aggregate is applied to concrete for a 21–27 MPa structure, the standard to facilitate the use of recycled fine aggregate has been prepared by revising 30% or less (using only recycled coarse aggregate) of the total aggregate capacity, as is also the case for non-structure concrete of less than 21 MPa. In the event that it is used for concrete for a structure of 27 MPa or less, the standard should be used 30% or less (using recycled coarse aggregate and recycled fine aggregate) of the total aggregate capacity, 60% or less (using only recycled coarse aggregate) of the total coarse aggregate capacity, and 30% or less (using only recycled fine aggregate) of the total fine aggregate capacity, and 30% or less (using only recycled coarse aggregate and recycled fine aggregate) of the total aggregate capacity. In addition, the standard value for recycled fine aggregates has been improved from the existing density of 2.2 g/cm$^3$ and absorption rate of 5%, to a density of 2.3 g/cm$^3$ and an absorption rate of 4% [21]. This was intended to minimize the quality deviation through an enhancement of quality for the recycled fine aggregate, but the current production technologies for recycled fine aggregates face difficulties in meeting the revised standard.

In this research, to use recycled aggregate in structural buildings, a facility was developed and research was conducted to remove the cement paste attached to the surface of the recycled aggregate, aggregate fine powder, and impurities. The characteristics of concrete using a recycled aggregate type and ratio were studied.

## 2. Scope and Method of Research

In this research, the quality improvement of the recycled fine aggregate and the characteristics of concrete using recycled aggregate have been studied through the following studies.

(1)　A study on the quality improvement of recycled fine aggregate

Recycled fine aggregate contains significant amounts of organic and inorganic impurities with an inconsistent quality, creating concerns about quality decline of the concrete. Therefore, the production facilities for quality improvement of the recycled fine aggregate were reviewed, and the equipment used to remove impurities and strip the recycled fine aggregate was developed for stable production and quality improvement of the recycled fine aggregate.

(2)　A study on the quality improvement of mixed aggregate

The recycled and natural aggregates have different specific gravities and are injected separately when producing concrete. If the mixer efficiency is low, this creates quality deviations in the concrete. Accordingly, to achieve a stable mixing of the recycled and natural aggregates, aggregate mixture facilities were developed. The densities, absorption rates, and other properties of the mixed aggregate, with various compositions were tested. In addition, the characteristics of the quality deviation of the recycled aggregate concrete, following premixing, were determined, and a premixing facility for the aggregate was developed.

(3)　An empirical study on the mechanical characteristics and durability of concrete using the recycled aggregate

Before and after modification, the recycled fine aggregates, recycled coarse aggregates, and natural aggregates that were mixed in accordance with the replacement rate of the physical and mechanical properties of the concrete that used the recycled fine aggregate, were evaluated. The concrete using the recycled aggregate was mixed with natural aggregate and the target strength was 24 MPa (general strength) or 35 MPa (high strength). Slump, air volume, compressive strength, drying shrinkage, freezing, and thawing resistance and other properties were measured to analyze the physical, mechanical and durability characteristics of the concrete containing recycled fine aggregate, before and after modification.

## 3. Development of the Technology to Improve the Recycled Fine Aggregate Quality

### 3.1. Outline

Waste materials collected from the construction fields are required by law to be classified, but due to the difficulty of field management, there is a significant amount of discharge of mixed waste materials. If the fine aggregate is less than 5 mm, it makes it difficult to separate the foreign matters on an aggregate surface. Therefore, there is a limit to the application of existing technologies for improving the density and absorption rate. In order to remove a large amount of impurities, particularly inflammable impurities, the most frequently used sorting method is separation by an air blower (as shown in Figure 2). Furthermore, the density and absorption rate of the recycled fine aggregate change depending on the contents of the mortar attached to the aggregate surface. As the mortar content increases, the density decreases and the absorption rate increases, diminishing the quality, as a result. Therefore, to improve the aggregate quality, the mortar attached to the aggregate surface must be removed as much as possible. In the past, there was no equipment available to remove foreign materials and mortar attached to the aggregate surface, at the same time.

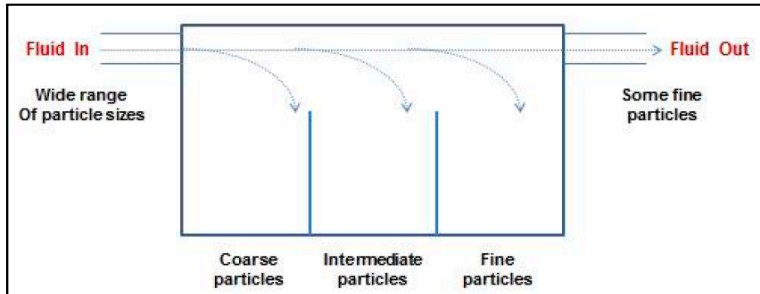

**Figure 2.** The principle of air-blower impurity sorting.

In general, employing the developed equipment, inflammable impurities with diameters >25 mm are separated through a separate sorting device, such as a trammel. Inflammable impurities in the 10–25 mm range are separated using a screen. Inflammable impurities of 10 mm or less are sorted by ventilation, during transport of the aggregate. Air-flow-type sorting methods have the advantages of being efficient in sorting impurities with low specific gravities; they also employ a simple facility configuration and equipment maintenance method. The purpose of separation selection should be clearly distinguished by locating at the end of processing. However, air-flow-type sorting, can sort out all substances with low specific gravities, including aggregates. Furthermore, this process can scatter well, and if there is a significant flow of aggregate, the sorting efficiency is low. Thus, a method for sorting impurities and for eliminating dust scattering is required, when there is a large amount of aggregates flowing.

Furthermore, the density and absorption rate of the recycled fine aggregate change, depending on the contents of the mortar attached to the aggregate surface. As the mortar content increases, the density decreases, and the absorption rate increases, diminishing the quality.

## 3.2. Principle of Technology

Technology is needed to resolve the quality decline, due to impurities, by processing the waste materials, implementing technology to remove mortar from the surface of the recycled fine aggregate, and producing recycled fine aggregate with a density of 2.3 g/cm$^3$ or more, an absorption rate of less than 4%, and organic impurity contents of less than 0.3%. The appearance and perspective drawing of the equipment for stripping impurities and recycled fine aggregates, and the entire production systems, are shown in Figures 3 and 4, respectively.

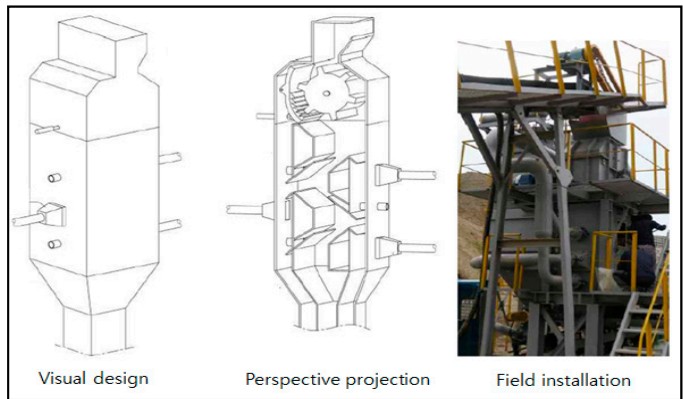

**Figure 3.** Appearance and perspective drawing of stripping the removal equipment for the impurities of recycled fine aggregate.

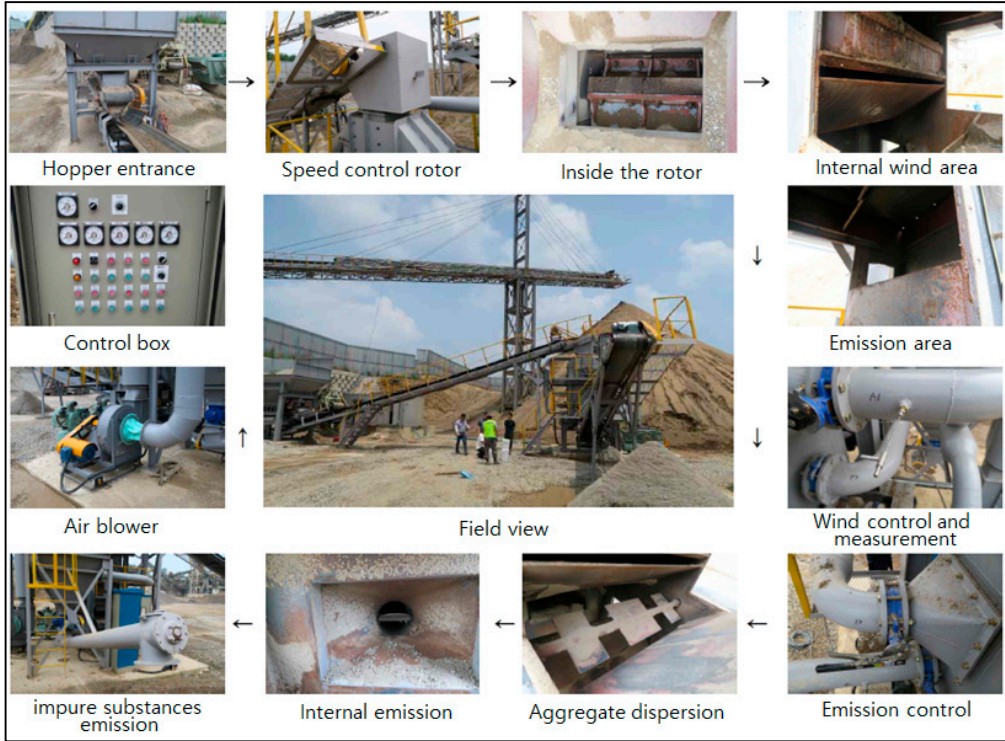

**Figure 4.** The production process of the technology development scheme.

These facilities are closed and can accommodate a multi-phase flow. The recycled fine aggregate is stripped by the Speed Adjusting Rotor Hammer, and the wind pressure and direction are varied, three times, to remove the impurities.

To separate the aggregate and impurities, a knife-type air-speed supply device and an inhaling device are used to capture the impurities. The wind pressure used to separate the impurities is generated by a recycling fan, with a capacity of 120 m$^3$/min and the structure between the upper rotation rotor and final discharge phase is enclosed to prevent the scattering of dust. In addition, by adjusting the wind velocity and absorption volume, the system can accommodate various impurities. Furthermore, the Speed Adjusting Rotor Hammer rotational impact device, accelerates the initially supplied aggregate, and the organic impurity is separated by the absorption device in the upper part, due to the wind speed of the rotational force. Mortar on the aggregate surface is removed through the stripping plate. The principle of mortar removal is shown schematically, in Figure 5.

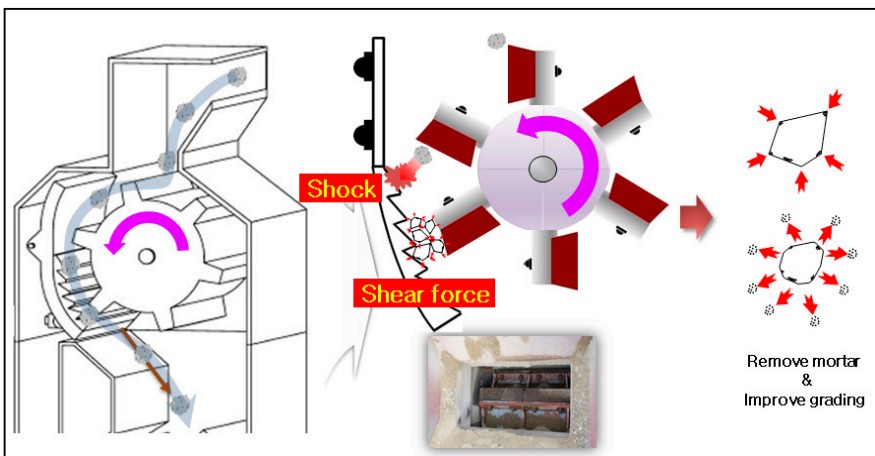

**Figure 5.** The principle of mortar removal.

*3.3. Experiments and Results of the Recycled Fine Aggregate Using Stripping Removal Equipment for the Impurities of Recycled Fine Aggregate*

The equipment developed in this study removed cement paste, aggregate particles, and the impurities attached to the surface of recycled aggregate. Removal of the cement paste increased the absolute dry density, decreased the absorption rate, and increased the percentage of the absolute volume of the particle shape. Additionally, through the removal of aggregate particles, the penetration rate of 0.08 mm was reduced and the content of organic and inorganic impurities was reduced through the removal of impurities.

### 3.3.1. Absolute Dry Density

The absolute dry density could affect the concrete strength if the aggregate density was lower than the Korean Standard requirement of 2.2 $g/cm^3$ or more. This test was implemented following the KS F 2504 "Method to Test Density and Absorption Rate of Fine Aggregate." The absolute dry density was found to increase by an average of approximately 0.07 $g/cm^3$. The dry density changed from 2.25 → 2.32 $g/cm^3$ in the first pass, to 2.25 → 2.33 $g/cm^3$ in the second pass, and 2.28 → 2.35 $g/cm^3$ in the third pass.

### 3.3.2. Absorption Rate

The recycled aggregate had significant amount of mortar attached to the aggregate surface, compared to the natural aggregate. Thus, the recycled aggregate had a greater absorption rate than that of the aggregate used for general concrete. If the recycled aggregate that had been air- or oven-dried was used, the water content of the aggregate was increased, to achieve more loss of slump in the unsolidified concrete, and the pumpability was worsened. Therefore, the KS required an absorption rate of 5% or less. This was measured following the KS F 2504 "Method to Test Density and Absorption Rate of Fine Aggregate." Before and after penetration, the absorption rate decreased by an average of approximately 0.70%. The absorption rate decreased from 5.42% → 4.62% in the first test, 5.35% → 4.68% in the second test, and 5.38% → 4.75% in the third test.

### 3.3.3. Percentage of Absolute Volume of the Particle Shape

The percentage of absolute volume of the particle shape was used to determine the aggregate particle shape. If the absolute volume percentage was 53% or more, the recycled fine aggregate particles were considered to be adequate. This was measured following the KS F 2527 "Aggregate Crushed for Concrete." Before and after penetration, the absorption rate increased by an average of approximately 0.64%. In the first, second, and third tests, the absolute volume percentage increased from 53.05% → 53.52%, 52.71% → 53.42%, and 52.88% → 53.63%, respectively.

### 3.3.4. Penetration Rate of 0.08 mm

In the event that there was a significant quantity of small particles with a size of 0.08 mm or less, the quality would be adversely influenced, resulting in, for example, a decline of strength, due to an increase of the unit quantity, when producing ready-mixed concrete. An increase of laitance and drying shrinkage when pouring the concrete, and a decrease in adhesion when joint pouring was applied, which was determined by the Korean Standard to be 7.0% or less. This test was implemented by the KS F 2511 "Method to test chips included in aggregate (penetrating the 0.08 mm sieve)", and as a result of the experiments before and after penetration, it was demonstrated that the chips decreased by an average of approximately 1.90% of 0.08 mm or less, in the first test, from 3.15% → 1.83%, in the second test from 3.65% → 1.27%, and in the third test from 3.39% → 1.39%. This was considered to be due to the air-flow removing the impurity of the development facility that sorted the chips.

### 3.3.5. Contents of Organic and Inorganic Impurities

Waste concrete has a variety of impurities that are mixed in the recycled aggregate, which was produced by crushing them, resulting in a declined quality. Accordingly, the KS F 2527 "Crushed Aggregate for Concrete", defined the impurity contents (organic and inorganic impurity) of the recycled fine aggregate. This test was implemented by the KS F 2576 "Method to Test the Foreign Substance Contents of Recycled Aggregates", and as a result of the experiment (as shown in Figure 6), before and after penetration, the organic impurity experiment demonstrated a decrease in the organic impurity, for an average of approximately 0.59%, when penetrating in the first test of 1.21% → 0.58%, the second test of 1.47% → 0.82%, and the third test of 1.25% → 0.77%, and as a result of the experiment, before and after penetration, in the inorganic impurity experiment, it demonstrated that a decrease in the contents of the inorganic impurity for an average of approximately 0.46%, in the first test, it showed a decrease from 0.95% → 0.48%, in the second test, from 1.05% → 0.55%, and in the third test, from 0.94% → 0.54%.

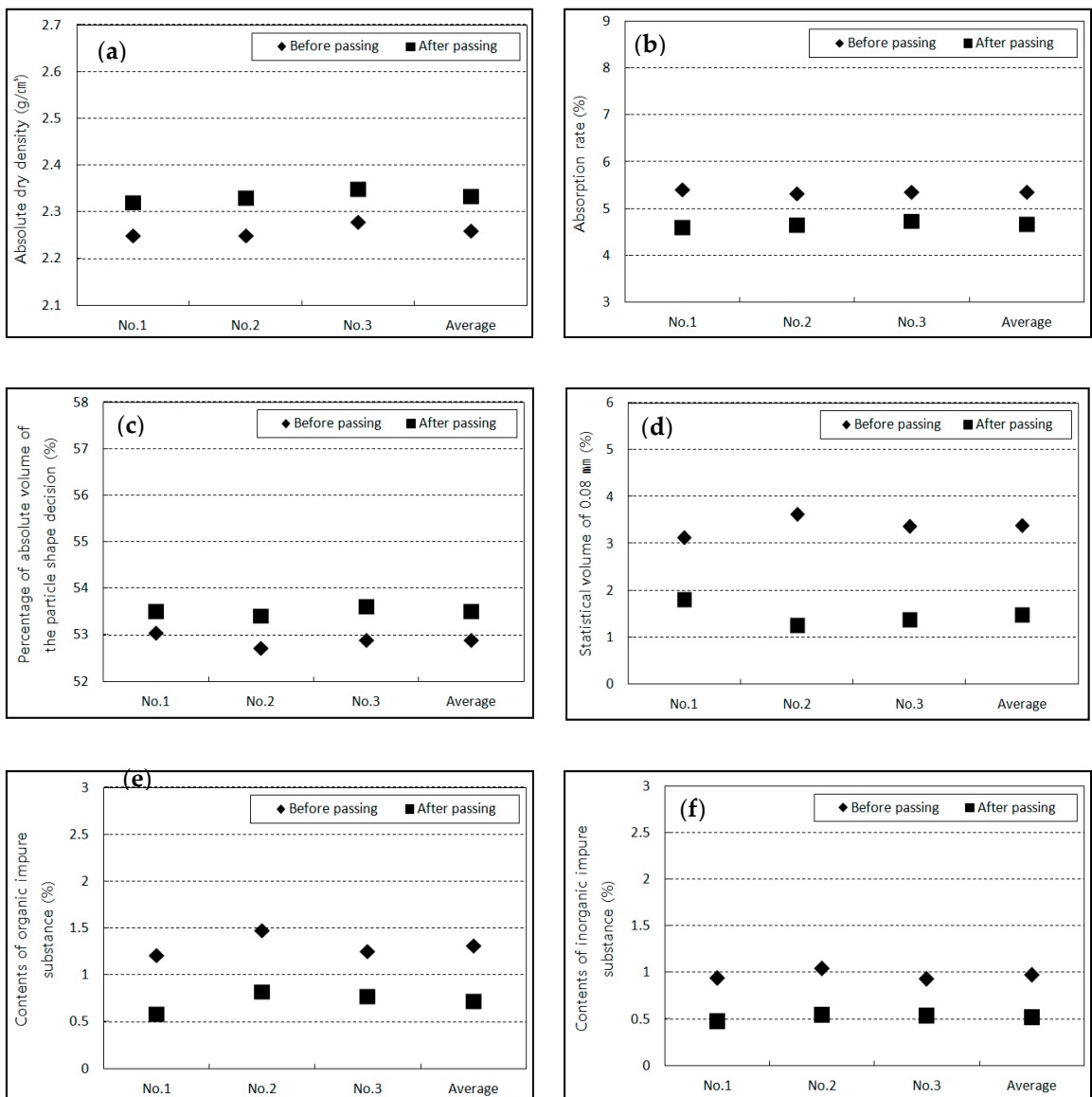

**Figure 6.** (**a**) Absolute dry density, (**b**) absorption rate, (**c**) percentage of absolute volume of the particle shape, (**d**) statistical volume of 0.08 mm, (**e**) contents of organic impurities, and (**f**) contents of inorganic impurities of recycled fine aggregate, before and after passing.

## 4. Development of the Technology to Improve the Quality of the Mixed Aggregates

### 4.1. Outline

Ready-mixed concrete is produced by weighing raw materials to inject into the mixer, at the time of production. However, the mixtures could differ depending on the mixer capability of the batch plant, and if the mixer efficiency is low, it could cause concrete quality deviations. In addition, if the materials with different densities are mixed by simultaneous injection, sufficient mixing is difficult due to the density difference. To improve the mixture capabilities, premixing is an option. According to the literature, if the mixture materials are displaced in a mass of approximately 70% or more, the density difference of cement, fine powder of the blast furnace slag, and fly ash could cause severe quality deviations. When the ready-mixed concrete production is conducted through advanced premixing, the mixing capabilities improve and the quality deviation is moderate [22].

### 4.2. Principle of Technology

In the Korea Standard, the material age is defined as the aggregate, with a density of 2.5 g/cm$^3$ or more, recycled coarse aggregate is defined as having a density of 2.5 g/cm$^3$ or more, and recycled fine aggregate is defined as having a density of 2.2 g/cm$^3$ or more. A quality deviation is generated, when mixing aggregates of different densities at a content of 70% or more, in the ready-mixed concrete. Technology to appropriately mix such mixtures was developed. The purpose of this was to produce recycled aggregate concrete with a difference of 0.8% or less, in the unit capacity volume of the mortar, and 5% or less in the unit coarse aggregate volume, using the hand mixing capability index defined by KS F 2455 "Test Method of Changes of Mortar and coarse aggregate volume from the Concrete Made with Mixer." The perspective drawing and mixing method of the aggregate mixing equipment are shown in Figure 7.

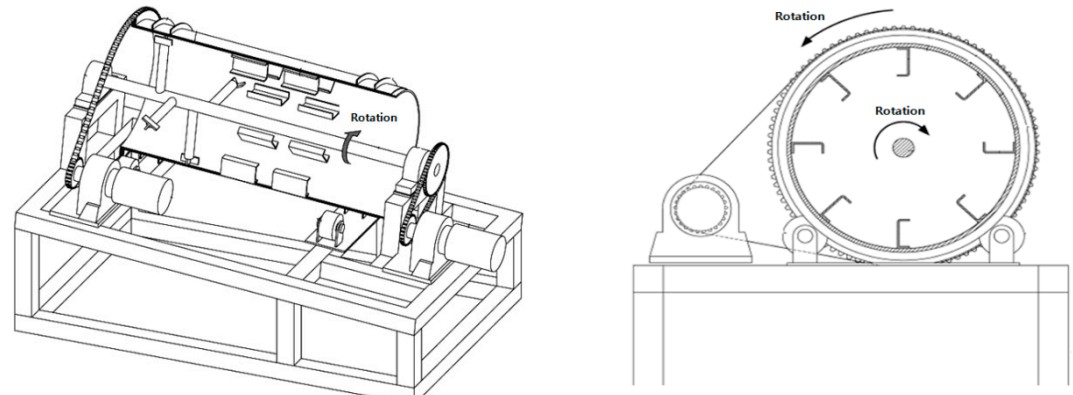

**Figure 7.** Perspective schematics of the aggregate mixing equipment and mixing method.

Previous aggregate mixing equipment have primarily used gravity methods. For conical-shaped mixers, the aggregate is rotated, allowing the cone to fall simply due to gravity. In this process, the aggregates fall in lumps, while they are attached to the inner wall of the cone-shaped mixer by centrifugal force. As a result, no mixing occurs.

This equipment uses gravity and forced mixing, with a mixing arm around the injection point, to mix the material as it is initially injected. A mixing plate is fixed inside the rotating drum, in a C-shape, for the secondary mixing, to prevent lumping when the material falls due to gravity. In addition, a counterclockwise rotation direction is used for a conical shape and a clockwise direction is used for the central axis. The mixing time is 30 s, and the mixing speed is 20 RPM.

*4.3. Experiment and Results of the Mixed Aggregate Quality before and after the Penetration of the Aggregate Mixing Equipment*

The mixed aggregate concrete before and after entering the aggregate mixing equipment was examined. The recycled coarse and fine aggregates were displaced by 50% of the entire aggregate volume with ready-mixed concrete after mixing for 45 s. The differences in the unit capacity volume of mortar in the concrete and differences in the unit coarse aggregate volume from the concrete, and the compressive strength under the Korea Standard, were investigated.

The experimental results are presented at Table 1.

**Table 1.** Physical properties of natural aggregate and recycled aggregate.

| Division | Unit Volume Weight (kg/m$^3$) | Absolute Dry Density (g/cm$^3$) | Absorption Rate (%) | Assembly Rate (%) | Organic Impurities (%) |
|---|---|---|---|---|---|
| Natural Coarse Aggregate | 1518 | 2.59 | 1.01 | 7.02 | - |
| Recycled Coarse Aggregate | 1485 | 2.51 | 5.85 | 6.26 | 0.53 |
| Natural Fine Aggregate | 1552 | 2.61 | 1.03 | 2.95 | - |
| Before Modified Recycled Fine Aggregate | 1305 | 2.21 | 4.89 | 2.48 | 0.92 |
| After Modified Recycled Fine Aggregate | 1384 | 2.33 | 3.62 | 2.74 | 0.49 |

4.3.1. Difference in Unit Capacity Volume of Mortar and Difference in Unit Coarse Aggregate Volume

The difference in the mortar unit capacity volume and the difference in the concrete before and after mixing are shown in Table 2. The difference in the unit capacity volume of mortar, before and after mixing was 16.17 kg at the front part and 15.84 kg at the rear part, for the case before mixing, demonstrating a difference of approximately 2.04%. After mixing, the unit capacity volume of mortar was 16.15 kg in the front part and 16.07 kg in the rear part, exhibiting a difference of approximately 0.49%. In addition, the difference in the unit coarse aggregate volume in the front part of 2.89 kg and the rear part of 2.69 kg was approximately 6.92% before mixing. After mixing, the front part was 2.87 kg and the rear part was 2.75 kg, representing a difference of 4.18%.

**Table 2.** Experimental results of Unit Volume Weight of Mortar and Unit Coarse Aggregate Amount.

| Division | Unit Volume Weight of Mortar | | | Unit Coarse Aggregate Amount | | |
|---|---|---|---|---|---|---|
| | Front (kg) | Back (kg) | Difference Rate (%) | Front (kg) | Back (kg) | Difference Rate (%) |
| Before | 16.17 | 15.84 | 2.04 | 2.89 | 2.69 | 6.92 |
| After | 16.15 | 16.07 | 0.49 | 2.87 | 2.75 | 4.18 |

4.3.2. Compressive Strength

This concrete compressive strength test was implemented to determine the difference in the compressive strength, before and after mixing (as shown in Figure 8). Before mixing and with a three-day material age, the front portion exhibited compressive strengths in the range of 11.1–12.5 Mpa, with an average of 11.8. In the back portion, the compressive strengths were in the range of 10.4–11.7 Mpa, with an average of 11.0 MPa. Thus, there was a difference in the compressive strengths of a 0.8 MPa between the front and back. After mixing, the front portion had a compressive strength range of 11.5–12.4 Mpa, for an average of 11.9 MPa. In the back portion, the range was 11.9–12.4 MPa, with an average of 12.2 MPa. Thus, there was a 0.3 MPa difference in the compressive strengths,

between the front and back. For a material with a seven-day age before mixing, the front portion had a range of 18.1–19.3 Mpa, with an average of 18.6 MPa. In the back portion, the range was 17.1–18.2 MPa, with an average of 17.8 MPa. Thus, there was a 0.8 MPa difference in the compressive strength, between the front and back. After mixing, the front portion had a range of 18.4–19.2 MPa, with an average of 18.8 MPa, and the back portion had a range of 17.0–19.2 MPa, with an average of 18.0 MPa.

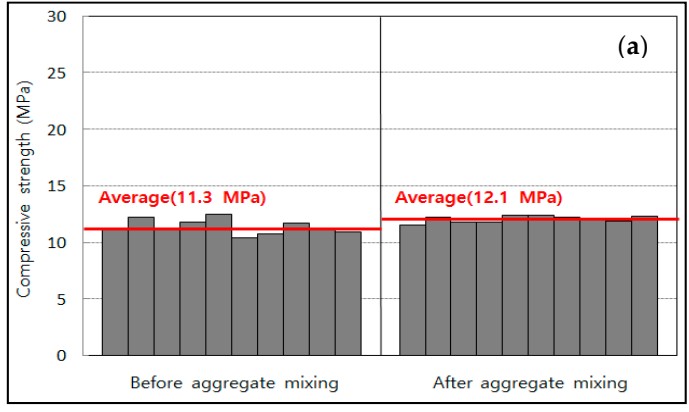

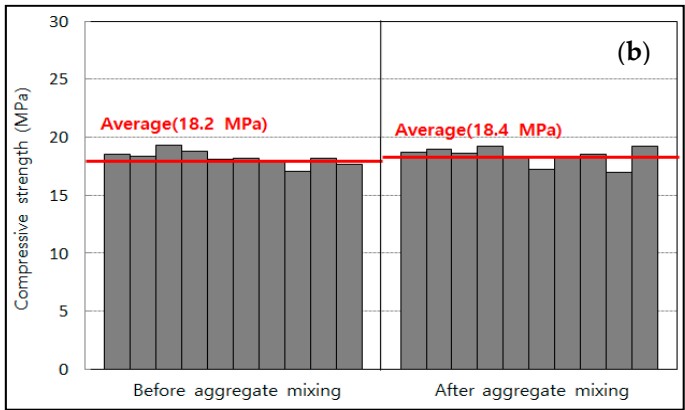

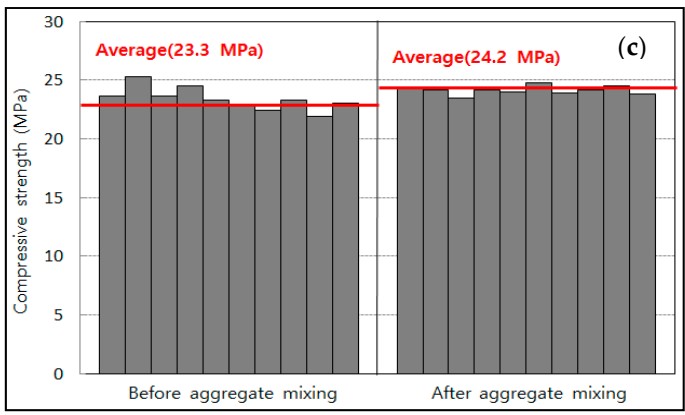

**Figure 8.** The (**a**) 3-day, (**b**) 7-day, and (**c**) 28-day concrete strength before and after aggregate mixing.

Thus, there was a 0.8 MPa difference in the compressive strengths between the front and back. For the material with a 28 day age before mixing, the front portion had a range of 23.3–25.3 Mpa, with an average of 24.1 MPa, and the back portion had a range of 21.9–23.3 Mpa, with an average of 22.7 MPa.

Thus, there was a 1.4 MPa difference in the compressive strengths between the front and back. After the mixture, the front portion had a range of 23.5–24.4 MPa, with an average of 24.1 MPa, and the back portion had a range of 23.8–24.8 MPa, with an average of 24.2 MPa. Thus, there was a 0.1 MPa difference in the compressive strengths between the front and back.

The differences in the compressive strengths between the samples with three and seven-day material ages had a range of approximately 1.1–1.4 Mpa, before mixing, and approximately 0.7–1.5 Mpa, after mixing. Thus, mixing decreased the difference. The compressive strength was approximately 0.8 Mpa, before mixing, and approximately 0.3 MPa after mixing. However, with a 28-day material age, the deviation of the compressive strength was in the range of approximately 1.4–2.0 Mpa, before mixing, and approximately 0.9–1.0 MPa after mixing, The compressive strength was approximately 1.4 Mpa, before mixing, and approximately 0.1 MPa after mixing, and there was a greater difference for the 28-day material age than for the three-day and seven-day material ages.

## 5. Mechanical Properties of the Recycled Aggregates Using Concrete and Durability Characteristics

### 5.1. Experimental Method

The purpose of this research was to expand the use of the recycled aggregate. The recycled aggregates, before and after modifications, were mixed in accordance with the replacement ratio, to evaluate the physical, mechanical, and durability properties of the recycled aggregate concrete. The slump, air volume, compressive strength, freezing and thawing resistance, drying shrinkage and other properties of the concrete, following the recycled aggregate replacement ratio were determined.

#### 5.1.1. Aggregate Replacement Rate

When the quality of the recycled fine aggregate was improved for use as a recycled aggregate, the recycled coarse aggregate was replaced by 0%, 30%, 60%, and 100%, for before and after modifications of the recycled fine aggregate, in order to find out the change in the capability of the recycled aggregate concrete.

#### 5.1.2. Experiment Factor, Level, and Combination Selection

The design standard strength had two levels, the general strength territory of 24 MPa and a high strength territory of 35 MPa, which were generally used for research. Experimental combinations were selected to satisfy the design standard strength, using the natural aggregate for the ready-mixed concrete combination in the preliminary experiment phase.

Concrete mix was determined so that the water to cement ratio (W/C) had a general strength territory of 44.3% and a high strength territory of 39.5%, sand to aggregate ratio (S/A) had a general strength territory of 47.1%, and a high strength territory of 48.0%.

An increase of the water-cement ratio for the recycled aggregate concrete tended to decrease the compressive strength. Similar to ordinary concrete, the aggregate quantity and water-cement ratio were fixed, and a water-reducing agent was used to achieve a target slump of 190 ± 25 mm. In addition, pre-wetting was implemented, prior to the experiment, to suppress the reduction of unit quantity by the air-gap on the recycled aggregate surface of the pastes, and the aggregates were implemented into the experiment under the internal saturation condition of the dried surface. The experiment factor and level of recycled aggregate concrete are shown in Table 3 and the experiment combinations are shown in Table 4.

**Table 3.** Experimental factors and levels.

| Division | Factors | Levels | Symbols |
|---|---|---|---|
| Aggregate Type | Recycled Coarse Aggregate, Recycled Fine Aggregate before Modify, Recycled Fine Aggregate after Modify | 3 | I, II, III |
| Recycled Aggregate Replacement Ratio (%) | 0, 30, 60, 100 | 4 | 1, 2, 3, 4 |
| Target Strength (MPa) | 24, 35 | 2 | A, B |
| Water Cement Ratio (%) | 48.5, 44.6 | 2 | - |
| Fine Aggregate Ratio (%) | 47.5, 49.2 | 2 | - |

**Table 4.** Concrete mixing table.

| Mix ID | W/C (%) | S/A (%) | Water | Cement | Sand N.F. * | Sand R.B. ** | Sand R.A. *** | Gravel N.C. **** | Gravel R.C. ***** |
|---|---|---|---|---|---|---|---|---|---|
| I1II1A | | | | | | | | 974.5 | 0 |
| I2II1A | | | | | 868.4 | 0 | | 682.2 | 292.4 |
| I3II1A | | | | | | | | 389.8 | 584.7 |
| I4II1A | | | | | | | 0 | 0 | 974.5 |
| I1II2A | 44.3 | 47.1 | 146.5 | 330.5 | 607.9 | 260.5 | | | |
| I1II3A | | | | | 347.4 | 521.0 | | | |
| I1II4A | | | | | 0 | 868.4 | | 974.5 | 0 |
| I1III2A | | | | | 607.9 | | 260.5 | | |
| I1III3A | | | | | 347.4 | | 521.0 | | |
| I1III4A | | | | | 0 | | 868.4 | | |
| I1II1B | | | | | | 0 | | 895.9 | |
| I2II1B | | | | | 825.6 | | | 627.1 | 268.8 |
| I3II1B | | | | | | | | 358.4 | 537.5 |
| I4II1B | | | | | | | 0 | 0 | 895.9 |
| I1II2B | 39.5 | 48.0 | 167.9 | 425.3 | 577.9 | 247.7 | | | |
| I1II3B | | | | | 330.2 | 495.4 | | | |
| I1II4B | | | | | 0 | 825.6 | | 895.9 | 0 |
| I1III2B | | | | | 577.9 | | 247.7 | | |
| I1III3B | | | | | 330.2 | 0 | 495.4 | | |
| I1III4B | | | | | 0 | | 825.6 | | |

* N.F.: Natural fine aggregate; ** R.B.: Recycled fine aggregate before modification; *** R.A.: Recycled fine aggregate after modification; **** N.C.: Natural coarse aggregate; and ***** R.C.: Recycled coarse aggregate.

## 5.2. Physical and Mechanical Characteristics

Air volume (as shown in Figure 9), slump (as shown in Figure 10), and compressive strength (as shown in Figure 11), test results were as follows.

### 5.2.1. Air Volume

The air volume increased in, both, the general strength territory and the high strength territory, as the replacement rate of the recycled coarse aggregate increased. The air volume increased slightly by 0.3%–0.5%, up to a replacement rate of 60%, which was not significant, compared to the air volume increase of ordinary concrete. However, at a replacement rate of 100%, the air volume increased by 1.1%–1.7%. In addition, during the replacement of the recycled fine aggregates, before modification, a replacement rate of 30% in the general strength territory and a high strength territory changed the air volume by −0.3%–0.1%, compared to the ordinary concrete. However, at a replacement rate of 60%, the air volume increased by 0.6%–1.6%. At a replacement rate of 100%, the air volume increased significantly by 2.0%–2.2%.

In the replacement of the recycled fine aggregate, after modification, both, the general strength territory and the high strength territory exhibited, decreased in air volume, −1.4% to −0.5%, compared to that of ordinary concrete, at up to 60% of the replacement rate. A similar air volume to that

of ordinary concrete was displayed at the replacement rate of 100%. Thus, the recycled aggregate contained a massive number of air-gaps on the mortar attached to the surface and the finite cracks, and the air volume increased as the density decreased, or the rate of use of the recycled aggregate (with a high absorption) rate increased. However, for the recycled fine aggregate, with an oven dried density of 2.33 g/cm$^3$ and an absorption rate of 4.62%, the air volume did not increase. Therefore, if the recycled coarse aggregate and the recycled fine aggregate, before modifications are used, it is prudent to prepare for changes in the air volume, for the design's standard strength and a continuous concrete quality management. Appropriate air volume management through an independent improvement of the recycled aggregate, is required.

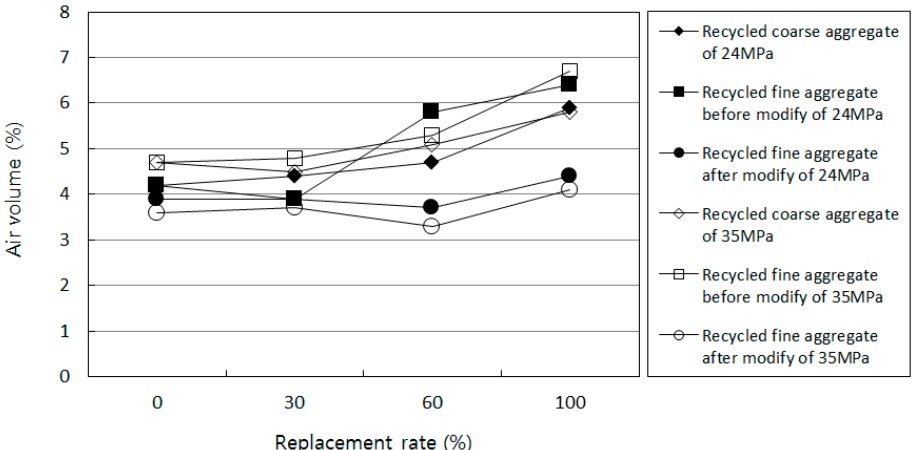

**Figure 9.** Air volume versus the replacement rate for each type of recycled aggregate for the general strength (24 MPa) and high strength (35 MPa) territory.

5.2.2. Slump

The elapse time of the slump showed that the slump decreased, following the elapse time, with an increase in the replacement rate for the general strength territory and a high strength territory, and up to 60% of the replacement rate. The slump decline was 55–75 mm at 60 min of elapse time, which was not significantly different to that of ordinary concrete at 60 mm. However, at 100% of the replacement rate, the slump decline was 100 mm at 60 min of elapse time. In addition, from the replacement of the recycled fine aggregate before modification, the general strength territory and high strength territory showed slump declines, following the elapse time with an increase in the replacement rate. Up to a 30% replacement rate, the slump decline was 60–70 mm, at 60 min elapse time, which was similar to the 60 mm of the slump decline of the ordinary concrete. However, at a 60% replacement rate, the slump decline was 80–105 mm, and at a 100% replacement rate, it was 115–150 mm. For the replacement of the recycled fine aggregate, after modification, up to 60% of the replacement rate, the slump decline was 50–55 mm at 60 min of elapse time, similar to that of ordinary concrete. At a 100% replacement rate, the slump decline was 60–75 mm, corresponding to a slump reduction of up to 20 mm, from ordinary concrete.

Thus, a replacement rate up to 60% for the recycled coarse aggregate, 30% for the recycled fine aggregate before modification, and 60% for the recycled fine aggregate after modification, were suitable. For any replacement of these limits, a slump reduction with elapse time occurred.

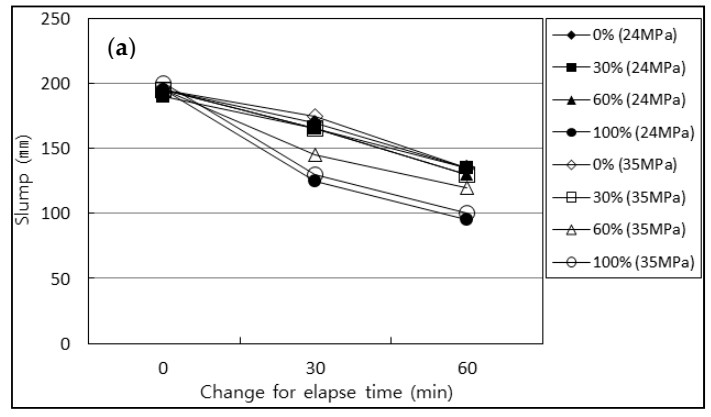

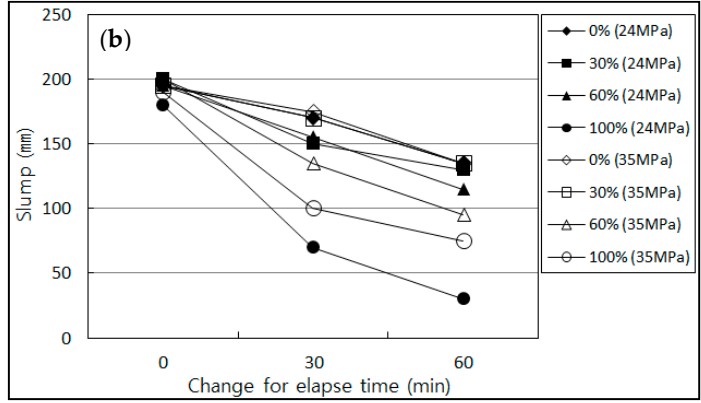

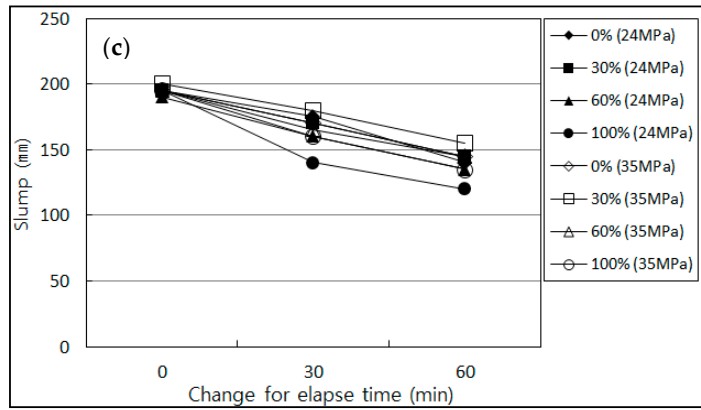

**Figure 10.** Slump change for strength area and replacement rate. (**a**) Recycled coarse aggregate replacement, (**b**) recycled fine aggregate before modification, and (**c**) recycled fine aggregate after modification.

### 5.2.3. Compressive Strength

Following the replacement of the recycled coarse aggregate and recycled fine aggregate, before and after modification, the compressive strengths were measured.

As the replacement rate of the recycled coarse aggregate increased, the compressive strength increased up to a 60% replacement rate, but drastically decreased at a 100% replacement rate. In particular, in the general strength territory, at a 60% replacement rate, the compressive strength was 26.3 MPa, which was 1.2 MPa greater than that of ordinary concrete (25.1 MPa). However, at a 100% replacement rate, the compressive strength was 19.8 MPa, which was 5.3 MPa less than that of ordinary concrete. In the high strength territory, at a 60% replacement rate, the compressive strength was 36.2 MPa, which was 0.3 MPa greater than that of ordinary concrete (35.9 MPa). However, at 100%,

the compressive strength was 32.2 MPa, which was 3.7 MPa less than that of ordinary concrete. This finding was similar to that of a previous study [23], in which the recycled coarse aggregate had a density of 2.51 g/cm$^3$ and an absorption rate of 2.85%. Furthermore, the aggregate was slightly stabilized so that the air volume following the replacement was minimal, up to a 60% replacement rate, and increased at 100%. Thus, in their study, there was no decline in the compressive strength for the recycled coarse aggregate, up to 60% of the replacement rate, but the decline of compressive strength was shown at 100% of the replacement rate.

Furthermore, the replacement rate of the recycled coarse aggregate had a strength improvement effect compared to ordinary concrete, by up to 60%, because the compressive strength improved with the increase of the absolute volume percentage in the concrete. The volume percentage increased because of the particle shape improvement, as a relatively round particle shape, with 3.93% of the natural coarse aggregate and 3.48% of the flattening rate of the recycled coarse aggregate. For the use of the recycled fine aggregate, before modification, the compressive strength decreased as the replacement rate increased, and at 30% of the replacement rate, there was a small reduction of 1.0 MPa in the general strength territory and 0.8 MPa in the high strength territory, compared to ordinary concrete. However, at 60% and 100% of the replacement rate, there was a slight decrease in the general strength territory of 3.8 and 8.3 Mpa and the high strength territory of 3.1 and 7.4 Mpa, compared to the ordinary concrete.

Based on previous studies, when the mortar attachment to the surface of the recycled fine aggregate was excessive, the replacement rate of the recycled fine aggregate was adequate, for 30% or less, but in excess of 30%. It was reported that the compressive strength decreased due to interference with the hydration products, following the excessive injection of the mortar particles, without reaction, due to the reduction effect of the water–cement ratio. Furthermore, in this study, up to 30% of the replacement of the recycled fine aggregate, before modification, produced minimal property changes, such as a reduction in the slump elapse time and increase of the air volume. However, 60% or more of the replacement produced property changes, and in particular, the air volume increased. This influenced the compressive strength [24,25]. When recycled fine aggregates after modification were used, at replacement rates of 0%, 30%, 60%, and 100%, the compressive strengths in the general strength territory were 25.7, 25.4, 26.5 and 23.8 MPa, respectively, and 36.1, 36.4, 37.3 and 34.8 MPa in the high strength territory, respectively. In particular, at the replacement rate of 100%, the general strength territory had a value of 1.9 MPa and the high strength territory had a value of 1.3 Mpa, compared to the ordinary concrete, showing a tendency of a slight decline, but the difference was minimal if the recycled fine aggregate with a density of 2.33 g/cm$^3$, absorption rate of 4.62%, and organic impurity content of 0.49% was used. This material could be used as an alternative material to natural aggregates.

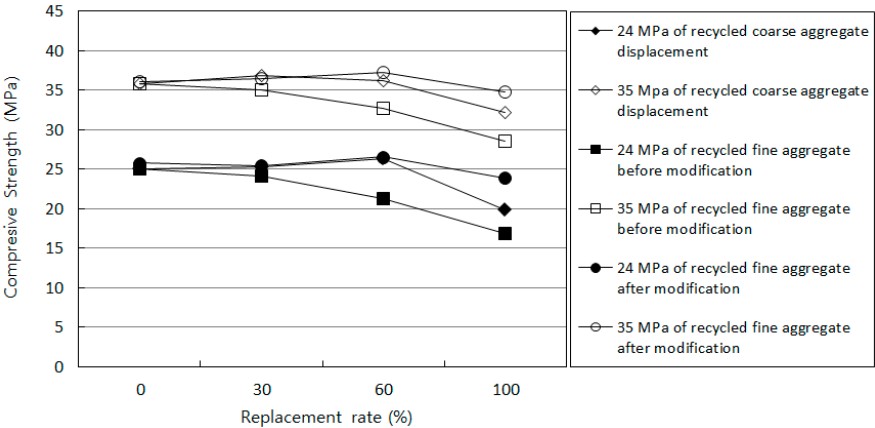

**Figure 11.** Compressive strength for strength area and replacement rate for recycled coarse aggregate replacement, recycled fine aggregate before modification, and recycled fine aggregate after modification.

*5.3. Durability Characteristics*

5.3.1. Freezing and Thawing Resistance

Freezing and thawing experiments of the recycled aggregate concrete were conducted according to KS F 2456 "standard test method for resistance of concrete to rapid freezing and thawing" and the experimental results are shown as Figure 12.

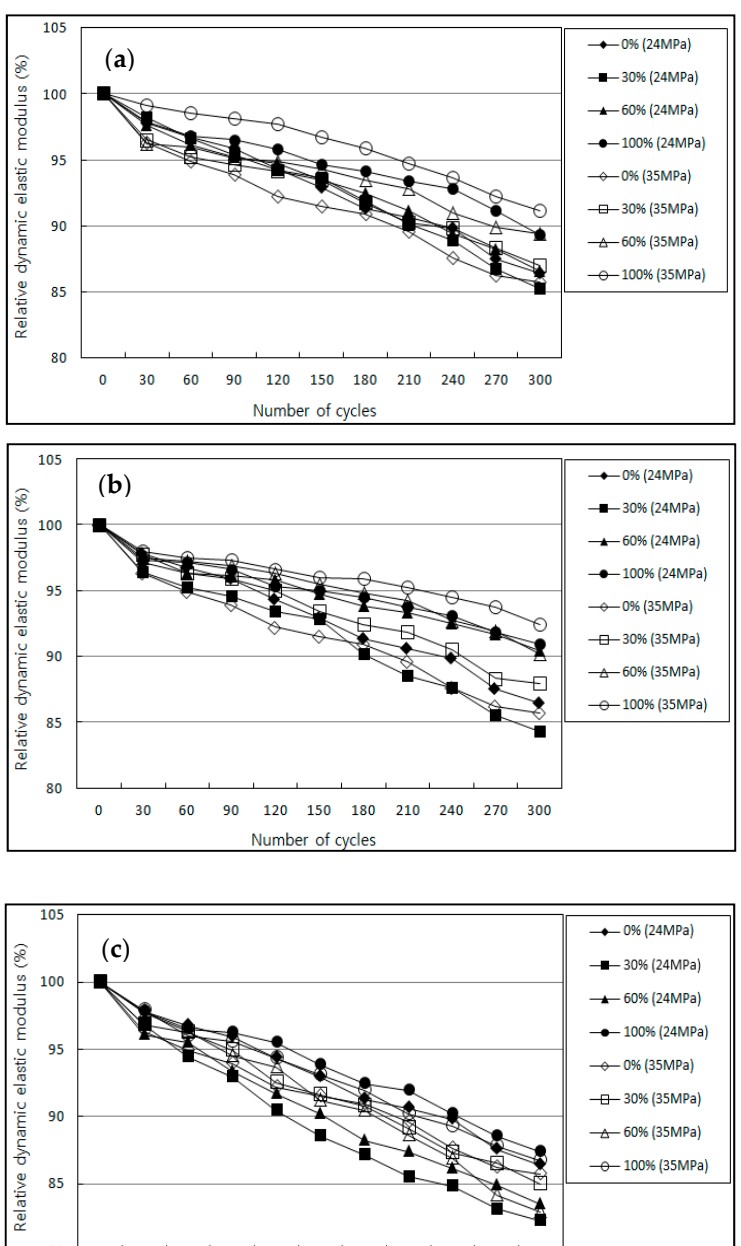

**Figure 12.** Relative dynamic elastic modulus and number of cycles: (**a**) Recycled coarse aggregate replacement, (**b**) recycled fine aggregate before modification, and (**c**) recycled fine aggregate after modification.

The durability index of ordinary concrete was 86, in the general strength territory and high strength territory. The durability indices of the recycled aggregate concrete that used the recycled coarse aggregate were 85, 87, and 89, in the general strength territory and 87, 89, and 91, in the high strength

territory at replacement rates of 30, 60, and 100%, respectively. The durability indices of the recycled aggregate concrete that used recycled fine aggregate after modification, exhibited slightly higher durability indices than that of ordinary concrete, with values of 84, 90, and 91, in the general strength territory and 88, 90, and 92, in the high strength territory for replacement rates of 30, 60, and 100%, respectively. Furthermore, in the recycled aggregate concrete that used the recycled fine aggregate after modification, the durability indices were 82, 84, and 87, in the general strength territory and 85, 83, and 87 in the high strength territory, for replacement rates of 30%, 60%, and 100%, respectively which were slightly lower than those of the recycled coarse aggregate and recycled fine aggregate, before modification. The durability index was thought to have increased due to a buffer effect, caused by the expansion and relaxation of moisture due to freezing and thawing. This is occurs due to a lowering of the air-spacing in the concrete as a result of the entrained air bubbles attached to the recycled aggregate. Furthermore, the amount of air entrained in the concrete containing recycled fine aggregate, after modification, is relatively small. In Figure 13, the relationship between the durability index and air volume of the recycled aggregate concrete is shown. As a result of the analysis, the significance probability was 0.856, so it was considered that there was a slightly high significance probability between the durability index and air volume of the recycled aggregate concrete. Accordingly, to improve the frost resistance, it was necessary to secure an appropriate air volume through combination management, through the use of an AE agent and other factors.

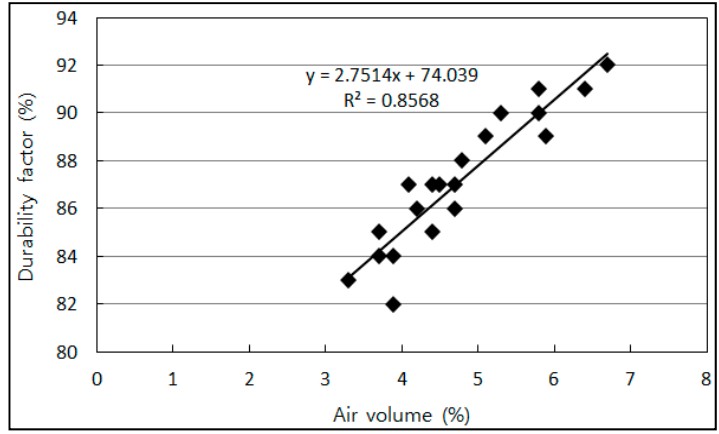

**Figure 13.** Relationship between durability and air volume.

### 5.3.2. Drying Shrinkage

Freezing and thawing experimental results are as Figure 14.

After 12 weeks of material aging, the length variation ratio due to the drying shrinkage of the ordinary concrete was $682 \times 10^{-6}$, in the general strength territory, and $526 \times 10^{-6}$, in the high strength territory. For the replacement rates of the recycled coarse aggregate of 30%, 60%, and 100%, the general strength territory exhibited length variations of 619, 652, and $533 \times 10^{-6}$, respectively, which were 4.4%–21.8% smaller than that of ordinary concrete. In the high strength territory, the length variations were 510, 476, and $508 \times 10^{-6}$, respectively, which were 3.0%–9.5% less than that of ordinary concrete. For the replacement rates of the recycled fine aggregate before modification of 30%, 60%, and 100%, the length variations in the general strength territory were 658, 417, and $454 \times 10^{-6}$, respectively, which were 3.5%–38.9% less than that of ordinary concrete, and in the high strength territory were 498, 371, and $324 \times 10^{-6}$, respectively, which were 5.3%–41.4% less than the ordinary concrete. In the replacement of the recycled fine aggregate after modification, for 30%, 60%, and 100% replacement rates, the length variations in the general strength territory were 697, 624, and $630 \times 10^{-6}$, respectively, which were $-2.2\%$–8.5% of that of ordinary concrete. The length variations in the high strength territory were 488, 497, and $424 \times 10^{-6}$, respectively, which were 5.5%–19.4% smaller than that of ordinary concrete.

Thus, when the recycled fine aggregate before modification was used, the length variations were similar to ordinary concrete, up to a replacement of 30%, but when the replacement was 60% and 100%, the length variation ratios were significantly reduced. After modification, the use of the recycled fine aggregate produced lower length variation ratios if the volume of the recycled aggregate was increased, but there was no significant decrease up to a difference of approximately 20%. Reducing the water–cement ratio reduced the amount of water absorbed into the air-gap of the mortar attached to the recycled aggregate.

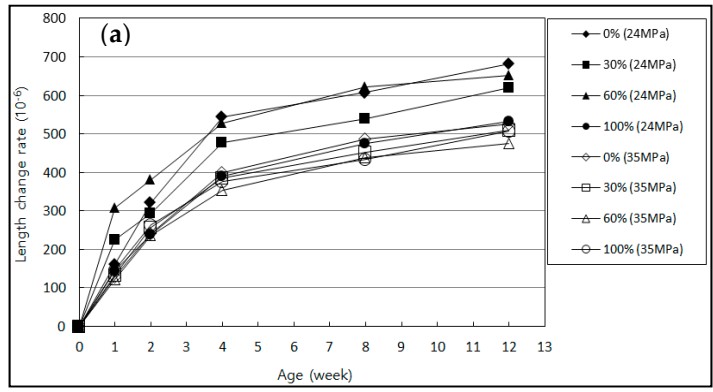

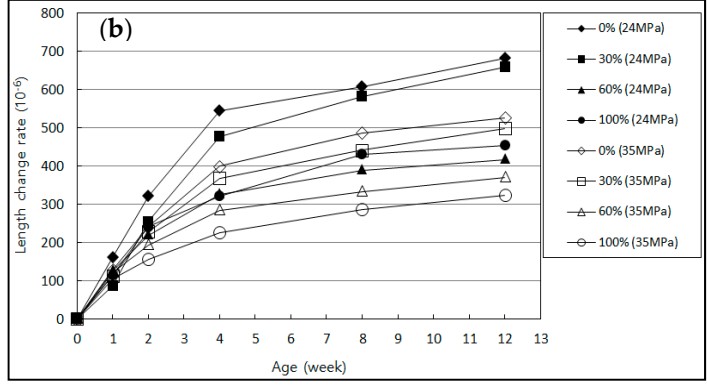

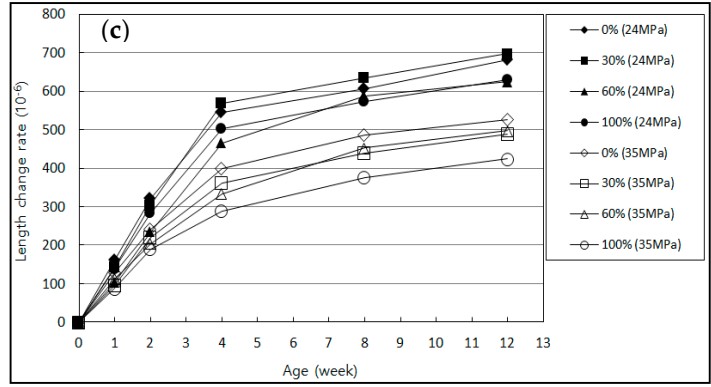

**Figure 14.** Length change rate according to age: (**a**) Recycled coarse aggregate replacement, (**b**) recycled fine aggregate before modification, and (**c**) recycled fine aggregate after modification.

## 6. Conclusions

The conclusions of this study are as follows.

- The quality before and after injecting the recycled fine aggregate using the equipment for impurity removal and stripping of recycled fine aggregate was examined. As a result, it was confirmed that the quality of the recycled fine aggregate was improved.
- For the recycled coarse aggregate, the increase in the air volume was minimal up to a replacement rate of 60%, but the air volume was increased by 1.1%–1.7% at a 100% replacement rate. For the recycled fine aggregate before modification, the air volume increased by 0.6%–1.6% at 60% of the replacement rate and 2.0%–2.2% at 100% of the replacement rate. However, there was no change in the air volume of the modified fine aggregate.
- For the recycled coarse aggregate in the slump tests, a replacement rate of 60% was appropriate. For the recycled coarse aggregate before modification, a 30% replacement rate was appropriate. For the recycled fine aggregate, a replacement rate of 60% was appropriate. For higher replacements, the slump decreased as the slump elapse time increased.
- For the recycled coarse aggregate, the compressive strength increased up to a replacement rate of 60% but decreased at a replacement rate of 100%. For the recycled fine aggregate before modification, as the replacement rate increased, the compression strength decreased. For the recycled fine aggregate after modification, a replacement rate of 100% produced similar results to those of ordinary concrete.
- Under the freezing and thawing resistance, the recycled coarse aggregate before modification and recycled fine aggregate before modification, exhibited slightly higher durability indices than that of the fine aggregate after modification. For the recycled fine aggregate after modification, the durability index was similar to that of ordinary concrete. In addition, in the mass reduction rate, the recycled coarse aggregate and the recycled fine aggregate after modification showed similar results to those of the ordinary concrete, for all combinations. However, the mass reduction rate of the recirculating fine aggregate, before modification, was increased from the replacement rate of 60% or more.
- If the recycled fine aggregate was used before modification in the drying shrinkage for up to 30% of replacement, the results were similar to those of ordinary concrete. However, for replacements of 60% or 100%, the length variation ratio significantly decreased. For the recycled coarse aggregate and the recycled fine aggregate after modification, the length variation ratio decreased when the volume of the recycled aggregate increased.

This study was conducted to generate the base data for the stable facilitation and expansion of recycled aggregate, through a concrete quality evaluation, by using the modified recycled fine aggregate and facility development for the stable production of recycled fine aggregate, under the revised quality standards. Recycled fine aggregate can be used with concrete, in an equal mixture, through aggregate pre-mixing before batch plant mixing, to improve the quality by using impurity removal and stripping removal equipment. Furthermore, this study is expected to contribute to the use of recycled aggregate in the future. As a follow-up study, the mechanical and durability characteristics of recycled aggregate concrete with varying density and various absorption rates should be explored and $CO_2$ emissions from recycled aggregate production should be studied.

**Author Contributions:** Conceptualization, J.-H.K.; Data curation, J.-H.K.; Formal analysis, J.-H.K., J.-H.S., C.-S.J. and S.-H.L.; Investigation, J.-H.S. and C.-S.J.; Project administration, J.-H.K., S.-H.L. and H.-S.K.; Supervision, H.-S.K.; Validation, H.-S.K.; Writing—original draft, J.-H.K.

**Funding:** This research was supported by a grant (18SCIP-C120606-03) from the Construction Technology Research Project Program funded by Ministry of Land, Infrastructure, and Transport of the Korean government.

**Conflicts of Interest:** The authors declare no conflict of interest.

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
