# Peer review of "A Study on the Properties of Recycled Aggregate Concrete and Its Production Facilities"

_applsci, doi:10.3390/app9091935_

Round 1

Reviewer 1 Report

This study introduces facilities for production high quality recycled aggregate and enhanced properties of concrete using modified aggregate. Before publication, following issues should be solved. 

1. There are lots of typos in the manuscript (For instance, 'increse' in the introduction). I encourage authors to go through the manuscript carefully. Moreover, English grammar and spelling should be edited extensively before resubmission. 

2. Introduction part is too long. It should be shortened. And author should clarify the goal of this research and the details of developed facilities.  

3. What is drying contraction? Do you mean drying shrinkage? 

4. In the section of 2.(3), what are other properties?

5. In 3.3, what is the meaning of penetration of impurities?

6. In figure 5, what is the meaning of before and after aggregate mixing? 'Before' means the results of cement paste?

7. In section 5.2.2, What is diurnal variation of the slump? Is it a normal term?

8. In references, some reference is phd thesis and the journals published in Korean journal. International journal papers should be referred. 

Author Response

Dear. Reviewer teacher

Thank you for your valuable comments. It helped me to improve the quality of thesis.

1. : There are lots of typos in the manuscript (For instance, 'increse' in the introduction). I encourage authors to go through the manuscript carefully. Moreover, English grammar and spelling should be edited extensively before resubmission.

→ Reply about 1. : We carefully reviewed the typo and modified it. when initial submission, We reviewed to Grammar proofreading company in order to the english grammar and spelling. If you need more grammar, we will proceed with an English review of the MDPI.

2. : Introduction part is too long. It should be shortened. And author should clarify the goal of this research and the details of developed facilities.

→ Reply about 2. : We reviewed the introduction part carefully and deleted a few of the parts that we thought were not important and we did clarify the goal of this research and the details of developed facilities.

3. : What is drying contraction? Do you mean drying shrinkage?

→ Reply about 3. : We corrected the word from drying contraction to drying shrinkage.

4. : In the section of 2.(3), what are other properties?

→ Reply about 4. : 2. (3) is to Evaluate the quality of before modify recycled fine aggregate and after modify recycled fine aggregate produced by the equipment developed in this study. As a result, we have studied the Mechanical and durability characteristics of recycled aggregate.

5. : In 3.3, what is the meaning of penetration of impurities?

→ Reply about 5. : It is the meaning of impurity removal which is not meaning of impurity penetration. Clearly, the subtitle has been modified. “ 3.3 Experiments and results of recycled fine aggregate using stripping removal equipment of impurity of recycled fine aggregate ”

6. : in figure 5, what is the meaning of before and after aggregate mixing? 'Before' means the results of cement paste?

→ Reply about 6. : The meaning before the aggregate mixing is not mixing the recycled aggregate and the natural aggregate, this is called "before". And the mixture of aggregate before mixing is called "After".

7. : In section 5.2.2, What is diurnal variation of the slump? Is it a normal term?

→ Reply about 7. : The words were changed to engineering words  “ elapse time ”

8. In references, some reference is phd thesis and the journals published in Korean journal. International journal papers should be referred.

→ Reply about 8. : In the introduction, the latest contents of the literature survey are attached and revised. - [REFERENCES] Slag waste incorporation in high early strength concrete as cement replacement: Environmental impact and influence on hydration; durability attributes, Early-age behavior of recycled aggregate concrete under steam curing regime, Removal of cement mortar remains from recycled aggregate using pre-soaking approaches, Resources, Conservation and Recycling, Properties of concrete prepared with PVA-impregnated recycled concrete aggregates, Cement and Concrete Composites, Use of a CO2 curing step to improve the properties of concrete prepared with recycled aggregates, Cement and Concrete Composites.

Thanks.

Reviewer 2 Report

This paper focuses the feasibility of producing high quality recycled concrete aggregates (RCA) for use in concrete for sustainable development. Strength and durability properties of concrete containing RCA were evaluated. Overall, the paper is an interesting read and possesses technical merits for its publication as a research article. However, there are several issues which must be addressed prior to its acceptance. Many key references are missing, as well as the quantitative measurement of improvement in environmental impact is not done. Determining the associated carbon dioxide emissions is highly imperative while advocating the use of RCA, which is not provided here.

Major and minor comments are appended below for Authors’ consideration.

1.     Line 15; delete space and correct the word “and”.

2.     Line 19; “after develop devices”…what do the authors mean? I suggest revising it for clarity.

3.     In the Abstract, the Authors talk of “environment improvement”, yet no quantification is done. In my opinion, determining the associated carbon dioxide emissions is highly essential. Please refer to the following papers in which the carbon dioxide emissions for recycled aggregate concrete (RAC) and normal concrete are calculated [1,2]. Please consider these citing appropriately while providing the details of CO2 emissions.

4.     Introduction needs thorough revision. The literature survey is not well up to date, currently. Many references pertaining to RCA improvement have been left out. This makes the Introduction very weak and not up to date. I highly recommend considering the following references to be included in the Introduction for updated literature review [3–8].

5.     The Figures captions must be corrected. Please use “Figure” instead of “Photo”.

6.     Table 1; use decimal points instead of commas.

7.     Another important issue is the lack of statistical information on the data analysis and results obtained. These include sample size, standard deviation / error, mean, etc. Please note that the reliability of the data can be corroborate only when ample information is provided. If there is high disparity among the different samples of the same mxi series can make the obtained results questionable. Also, the error bars must be included in the Figures which refer to resulting concrete properties.

8.     Table 4; define “S/A”.

9.      Figure 6; the x-axis caption should be “Replacement Ratio (%)”. Pease correct. Also please double check the figure legend.

10.  Section 5.3.2 is related to “Shrinkage”.  Please use the correct technical term.

11.  It is also advisable to compare the obtained results of strength and shrinkage with the previously published findings to indicate the better performance of RAC produced in this study.

12.  Conclusions are merely repeating the results and discussion part. Please revise the Conclusions and include the future recommendation, challenges and applicability of the research findings.

REFERENCES

[1]      Y. Kim, A. Hanif, M. Usman, M.J. Munir, S.M.S. Kazmi, S. Kim, Slag waste incorporation in high early strength concrete as cement replacement: Environmental impact and influence on hydration; durability attributes, Journal of Cleaner Production. 172 (2018) 3056–3065. doi:10.1016/j.jclepro.2017.11.105.

[2]      A. Hanif, Y. Kim, Z. Lu, C. Park, Early-age behavior of recycled aggregate concrete under steam curing regime, Journal of Cleaner Production. 152 (2017) 103–114. doi:10.1016/j.jclepro.2017.03.107.

[3]      V.W.Y. Tam, C.M. Tam, K.N. Le, Removal of cement mortar remains from recycled aggregate using pre-soaking approaches, Resources, Conservation and Recycling. 50 (2007) 82–101. doi:10.1016/j.resconrec.2006.05.012.

[4]      S.C. Kou, C.S. Poon, Properties of concrete prepared with PVA-impregnated recycled concrete aggregates, Cement and Concrete Composites. 32 (2010) 649–654. doi:10.1016/j.cemconcomp.2010.05.003.

[5]      S.C. Kou, B.J. Zhan, C.S. Poon, Use of a CO2 curing step to improve the properties of concrete prepared with recycled aggregates, Cement and Concrete Composites. 45 (2014) 22–28. doi:10.1016/j.cemconcomp.2013.09.008.

Author Response

Dear. Reviewer teacher

Thank you for your valuable comments. It helped me to improve the quality of thesis.

1. : Line 15; delete space and correct the word “and”.

→ Reply about 1. : I modified line 15 to fit the form and corrected the word "and".

2. : Line 19; “after develop devices”…what do the authors mean? I suggest revising it for clarity.

→ Reply about 2. : Device development is clearly modified. “ after develop equipment to improve the quality of recycled aggregate ”

3. : In the Abstract, the Authors talk of “environment improvement”, yet no quantification is done. In my opinion, determining the associated carbon dioxide emissions is highly essential. Please refer to the following papers in which the carbon dioxide emissions for recycled aggregate concrete (RAC) and normal concrete are calculated [1,2]. Please consider these citing appropriately while providing the details of CO2 emissions.

→ Reply about 3. : I did not know that CO2 emissions were important during my research. Sorry about that. A follow-up study will be conducted on the emission of CO2 emissions. And we will study the CO2 emission by studying the previous study about CO2 emission in the future.

4. : Introduction needs thorough revision. The literature survey is not well up to date, currently. Many references pertaining to RCA improvement have been left out. This makes the Introduction very weak and not up to date. I highly recommend considering the following references to be included in the Introduction for updated literature review [3–8].

→ Reply about 4. : In the introduction, the latest contents of the literature survey are attached and revised. - [REFERENCES] Slag waste incorporation in high early strength concrete as cement replacement: Environmental impact and influence on hydration; durability attributes, Early-age behavior of recycled aggregate concrete under steam curing regime, Removal of cement mortar remains from recycled aggregate using pre-soaking approaches, Resources, Conservation and Recycling, Properties of concrete prepared with PVA-impregnated recycled concrete aggregates, Cement and Concrete Composites, Use of a CO2 curing step to improve the properties of concrete prepared with recycled aggregates, Cement and Concrete Composites.

5. : The Figures captions must be corrected. Please use “Figure” instead of “Photo”.

→ Reply about 5. : As the reviewer said, I have modified from photo to figure.

6. : Table 1; use decimal points instead of commas.

→ Reply about 6. : In Table 1, the unit volume weight (kg / m3) of 1,518 / 1,485 / 1,552 / 1,384 was modified to 1 518/1 485/1 552/1 384 to fit the international unit.

7. : Another important issue is the lack of statistical information on the data analysis and results obtained. These include sample size, standard deviation / error, mean, etc. Please note that the reliability of the data can be corroborate only when ample information is provided. If there is high disparity among the different samples of the same mxi series can make the obtained results questionable. Also, the error bars must be included in the Figures which refer to resulting concrete properties.

→ Reply about 7. : The results of this study are to examine deviations from the compressive strength measurements. In general, there is no error range when measuring compressive strength and it shows various compressive strength results. This is why we want to analyze the differences in the various compressive strengths.

8. Table 4; define “S/A”.

→ Reply about 8. : I have defined S/A. " 5.1.2. Experiment factor, level and combination selection / Concrete mix was determined that W/C (water ratio to cement) is general strength territory 44.3%, high strength territory 39.5%, S/A (sand ratio to aggregate) is general strength territory 47.1%, high strength territory 48.0%. "

9. : Figure 6; the x-axis caption should be “Replacement Ratio (%)”. Pease correct. Also please double check the figure legend.

→ Reply about 9. : I changed the contents and pictures from displacement to replacement.

10. : Section 5.3.2 is related to “Shrinkage”. Please use the correct technical term.

→ Reply about 10. :I used accurate technical terms with shrinkage.

11. : It is also advisable to compare the obtained results of strength and shrinkage with the previously published findings to indicate the better performance of RAC produced in this study.

→ Reply about 11. : Compared with previous studies. It was found that the recycled fine aggregate before modification with a large amount of cement paste exhibited low performance in slump and air volume, and the compressive strength was influenced by air volume. If recycled fine aggregate is used before modification, countermeasures are needed.

12. : Conclusions are merely repeating the results and discussion part. Please revise the Conclusions and include the future recommendation, challenges and applicability of the research findings.

→ Reply about 12. : We modified the conclusion. As a follow-up study, it is necessary to study the mechanical and durability characteristics of concrete using recycled aggregates with various densities and various absorption rates. and It is considered that quantitative evaluation study of CO2 emissions should be carried out in the improvement of recycled aggregate.

Thanks.

Round 2

Reviewer 1 Report

The authors revised the manuscript well in accordance with the reviewers' comments. 

However, there are still English grammar issue. Please go through the manuscript once more before the final publication. Here are some minor comments. 

Photo-> Figure. 

    2. I am still confused with before and after the mixing.... 

Author Response

Dear. Reviewer teacher

Thank you for your valuable comments. It helped me to improve the quality ofthesis.

1. :  There are still English grammar issue. Please go through the manuscript once more before the final publication. Here are some minor comments.

 â†’Reply about 1. : My paper modified English grammar issue in the mdpi editing department

2. :  I am still confused with before and after the mixing. 

→ Reply about 2. : The meanings before and after mixing are as follows.

   “After mixing” : Aggregates mixed using an aggregate mixing plant

   “Before mixing” : Aggregates without aggregate mixing plant

Thanks.

Reviewer 2 Report

I have reviewed the manuscript. The Authors have addressed fairly to the Reviewers’ comments. Certain comments were left out and the paper is still not considered up to the mark. Following minor amendments are further needed to improve the manuscript.

1.     References are still inadequate in the Introduction. As the paper is focusing the pretreatment of recycled aggregates, in my opinion pretreatment-related paper should be cited. I had also mentioned this in the first review round. Please consider citing the below mentioned article in Introduction.

(a). Y. Kim et al., Properties enhancement of recycled aggregate concrete through pretreatment of coarse aggregates – Comparative assessment of assorted techniques, Journal of Cleaner Production. 191 (2018) 339–349. doi:10.1016/j.jclepro.2018.04.192.

2.     Also, Discussion must include the primary reason of strength decline, which is “porosity’. Please have a look at the following papers. These should be appropriately cited, I think in section 5.2.1. The goal is to make the readers understand.

(a). Y. Kim, et al., Influence of bonded mortar of recycled concrete aggregates on interfacial characteristics – Porosity assessment based on pore segmentation from backscattered electron image analysis, Construction and Building Materials. 212 (2019) 149–163. doi:10.1016/j.conbuildmat.2019.03.265.

(b).  P.W. Barnhouse and W. V. Srubar, Material characterization and hydraulic conductivity modeling of macroporous recycled-aggregate pervious concrete, Construction and Building Materials. 110 (2016) 89–97. doi:10.1016/j.conbuildmat.2016.02.014.

3.     As mentioned in the first review round, the use of “photo” should be avoided. It’s unprofessional. Please use “Figure” instead.

4.     Figure 11, error bars?

Author Response

Dear. Reviewer teacher

Thank you for your valuable comments. It helped me to improve the quality of thesis.

And my paper modified English grammar issue in the mdpi editing department

1. : References are still inadequate in the Introduction. As the paper is focusing the pretreatment of recycled aggregates, in my opinion pretreatment-related paper should be cited. I had also mentioned this in the first review round. Please consider citing the below mentioned article in Introduction.

→Reply about 1. : Kim, Hanif, Kazmi, Munir, and Park used two methods for pretreating aggregate. The first is HCl pretreatment and the second is Na2SO4 pretreatment. As a result, the overall concrete quality was improved with an increase in compressive strength of 14%.

 And I cited this study in my paper

2. : Discussion must include the primary reason of strength decline, which is “porosity’. Please have a look at the following papers. These should be appropriately cited, I think in section 5.2.1. The goal is to make the readers understand.

→Reply about 2. : I cited the reviewer's opinion that the compressive strength is reduced by the effect of air volume on 5.2.3 "compressive strength".

3. : As mentioned in the first review round, the use of “photo” should be avoided. It’s unprofessional. Please use “Figure” instead.

→Reply about 3. : I modified from photo to figure.

4. : Figure 11, error bars?

→Reply about 3. : Fugure 11 is not error.

Thanks